# Cryo-EM structures of thylakoid-located voltage-dependent chloride channel VCCN1

Tatsuya Hagino[1], Takafumi Kato[1,8], Go Kasuya [2], Kan Kobayashi[1,9], Tsukasa Kusakizako [1], Shin Hamamoto[3,4], Tomoaki Sobajima [5], Yuichiro Fujiwara[6], Keitaro Yamashita [1,10], Hisashi Kawasaki[3,4], Andrés D. Maturana[7], Tomohiro Nishizawa [1,11✉] & Osamu Nureki [1✉]

In the light reaction of plant photosynthesis, modulation of electron transport chain reactions is important to maintain the efficiency of photosynthesis under a broad range of light intensities. VCCN1 was recently identified as a voltage-gated chloride channel residing in the thylakoid membrane, where it plays a key role in photoreaction tuning to avoid the generation of reactive oxygen species (ROS). Here, we present the cryo-EM structures of *Malus domestica* VCCN1 (MdVCCN1) in nanodiscs and detergent at 2.7 Å and 3.0 Å resolutions, respectively, and the structure-based electrophysiological analyses. VCCN1 structurally resembles its animal homolog, bestrophin, a $Ca^{2+}$-gated anion channel. However, unlike bestrophin channels, VCCN1 lacks the $Ca^{2+}$-binding motif but instead contains an N-terminal charged helix that is anchored to the lipid membrane through an additional amphipathic helix. Electrophysiological experiments demonstrate that these structural elements are essential for the channel activity, thus revealing the distinct activation mechanism of VCCN1.

[1] Department of Biological Science, Graduate School of Science, The University of Tokyo, Tokyo, Japan. [2] Division of Integrative Physiology, Department of Physiology, Jichi Medical University, Shimotsuke, Japan. [3] Agro-Biotechnology Research Center, Graduate School of Agricultural and Life Sciences, The University of Tokyo, Tokyo, Japan. [4] Collaborative Research Institute for Innovative Microbiology, The University of Tokyo, Tokyo, Japan. [5] Department of Biochemistry, University of Oxford, Oxford, UK. [6] Molecular Physiology and Biophysics, Faculty of Medicine, Kagawa University, Miki, Japan. [7] Department of Applied Biosciences, Graduate School of Bioagricultural Sciences, Nagoya University, Nagoya, Japan. [8] Present address: Department of Biochemistry, University of Oxford, Oxford, UK. [9] Present address: Peptidream Inc, Kawasaki, Japan. [10] Present address: Structural Studies Division, MRC Laboratory of Molecular Biology, Cambridge, UK. [11] Present address: Graduate School of Medical Life Science, Yokohama City University, Yokohama, Japan. ✉email: t-2438@yokohama-cu.ac.jp; nureki@bs.s.u-tokyo.ac.jp

Photosynthesis by plant cells is a crucial metabolic reaction that generates oxygen and carbohydrates. In this reaction, light energy is absorbed and converted into chemical molecules, including ATP, at the thylakoid membrane of chloroplasts[1]. Absorbed light energy drives water oxidation and $H^+$ transport by the photosystem II (PSII) and cytochrome $b_6f$ complex, to generate the proton motive force (PMF) across the membrane[2]. The PMF consists of the potential gradient ($\Delta\Psi$) and the $H^+$ gradient ($\Delta pH$), and promotes ATP synthesis[3]. Although light-induced PMF generation is necessary, excessive photoreaction induces the generation of reactive oxygen species (ROS), which cause serious damage to chloroplasts[4]. Therefore, plants have adopted various regulation systems to control excess photon production. For example, non-photochemical quenching (NPQ) at the thylakoid membrane, which is widely conserved among plants, dissipates excessive light energy as heat[5]. The acidic luminal environment formed by the $H^+$ uptake enhances NPQ, and therefore the PMF is mainly stored as $\Delta pH$ to activate NPQ and allow the rapid adjustment of photosynthesis against light intensity changes[6]. To maintain the high contribution of $\Delta pH$, $\Delta\Psi$ is dissipated by ion counterbalancing across the thylakoid membrane, via cation efflux and anion influx by various membrane transporters and channels[6,7]. The thylakoid-located two-pore $K^+$ channel, TPK3, functions in cation efflux[8]. For NPQ downregulation, the $K^+/H^+$ antiporter KEA3 reportedly effluxes $H^+$ to adjust photosynthesis for the transition from high to low light[9,10]. As for anion influx, patch-clamp studies suggested the existence of a voltage-gated anion channel in the thylakoid membrane, which could play a key role in rapid $Cl^-$ influx in response to the $\Delta\Psi$ change caused by $H^+$ uptake[11–13]. VCCN1, the voltage-gated $Cl^-$ channel localized in the thylakoid membrane, was identified by mutant screening using a mutagenic alkylating agent (Ethyl Methanesulfonate; EMS) according to the diminished NPQ[14]. Further mutant analyses revealed that VCCN1 plays a key role in partitioning the PMF and contributes to the fine-tuning of photosynthesis in the transition from low to high light[14–16] (Supplementary Fig. 1).

VCCN1 belongs to the bestrophin family, which is widely conserved among kingdoms. In animal cells, bestrophin channels function as $Ca^{2+}$-activated $Cl^-$ channels, and mutations of BEST1, the most functionally characterized protein among the bestrophin channels, are associated with retinal diseases, including vitelliform macular dystrophy[17]. The structures of bestrophin channels and their bacterial homologs are well characterized: both form a homopentamer with each subunit containing four trans-membrane helices[18,19]. Furthermore, bestrophin channels are gated by $Ca^{2+}$-binding to the conserved motif termed the "$Ca^{2+}$ clasp" and by a long C-terminal inactivation peptide[19,20]. A previous electrophysiological experiment using cell-free expressed AtVCCN1 proteins reconstituted in planar lipid bilayers revealed voltage-sensitive anion channel activity, suggesting that VCCN1 is predominantly regulated by the membrane voltage[14,15]. Accordingly, VCCN1 has low amino acid sequence similarity with bestrophin channels — 9% sequence identity between *Arabidopsis thaliana* VCCN1 (AtVCCN1) and human Bestrophin 1 (HsBEST1) — and is predicted to lack both the $Ca^{2+}$ clasp and long C-terminal petide[14,15]. Instead, VCCN1 has a unique N-terminal sequence that is not present in bestrophin channels. These differences between the animal and plant homologs suggest that the regulatory mechanism of VCCN1 is distinct from those of bestrophin channels.

In this work, we present the cryo-EM structures of the apple VCCN1 in lipid nanodiscs and in detergent micelles. Our structures, together with electrophysiological experiments, provide insights into the channel properties and gating mechanism of VCCN1.

## Results

**Functional characterization and structure determination of MdVCCN1.** We first created various constructs of plant VCCN1 homologs with truncations of the N-terminal chloroplast targeting peptides, and screened them by fluorescence detection size-exclusion chromatography (FSEC)[21]. One of the truncated constructs from the apple VCCN1 protein (MdVCCN1; "Md" refers to *Malus domestica*, and hereafter referred to as the "wild-type" construct), which shares 73% sequence identity with AtVCCN1 (Supplementary Fig. 2), showed a monodisperse FSEC profile with high expression. We next assessed the functional properties of MdVCCN1 by whole-cell patch-clamp recordings. HEK293 cells expressing MdVCCN1 showed outwardly rectifying currents (Fig. 1a), and the channel activity was diminished by the addition of a $Cl^-$ channel inhibitor, 4,4'-diisothiocyanato-2,2'-stilbenedisulfonic acid; DIDS (Fig. 1b, c), consistent with the previous experiment using the AtVCCN1 proteins reconstituted in a lipid environment[15].

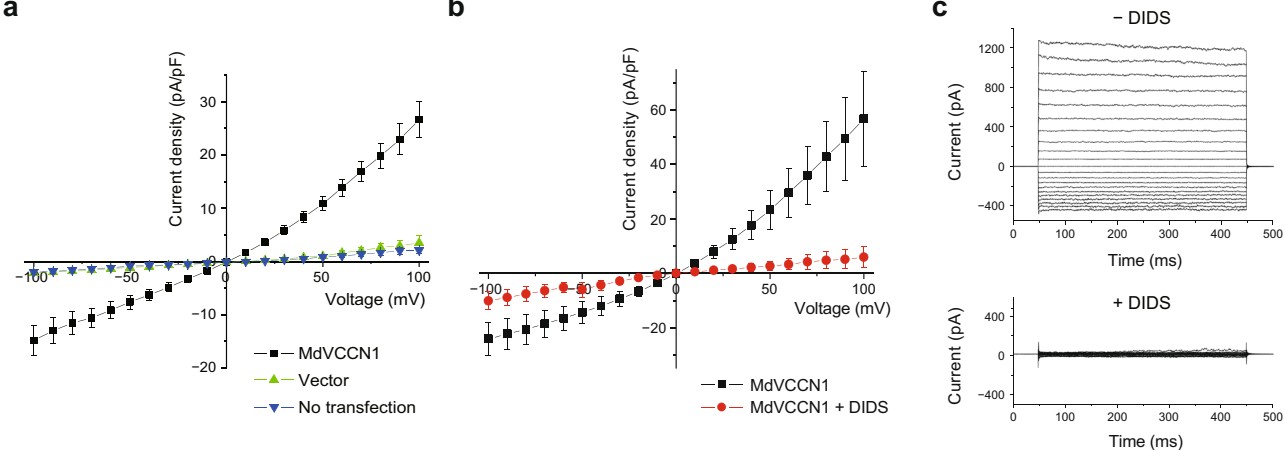

**Fig. 1 Functional characterization of MdVCCN1. a** Whole-cell current-voltage relationships in HEK293 cells expressing MdVCCN1 ($n = 7$), expressing GFP by transfection of the empty GFP co-expression vector ($n = 10$), and non-transfected HEK293 cells ($n = 9$). Error bars represent s.e.m. Source data are provided as a Source Data file. **b** Whole-cell current-voltage relationships in HEK293 cells expressing MdVCCN1, before (black) and after (red) the addition of 0.5 mM DIDS ($n = 8$). Error bars represent s.e.m. Source data are provided as a Source Data file. **c** Representative traces of whole-cell currents in HEK cells expressing MdVCCN1, before and after the addition of 0.5 mM DIDS to the bath solution. Source data are provided as a Source Data file.

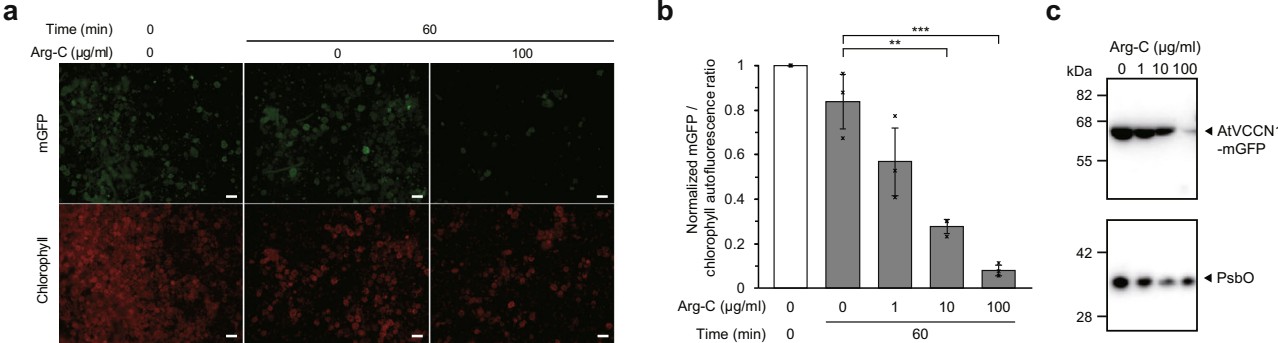

**Fig. 2 Topology evaluation of VCCN1 in chloroplasts. a** Representative images of mGFP digestion by Arg-C on isolated chloroplasts with transient expression of AtVCCN1-mGFP. The scale bar represents 50 μm. **b** Quenching level of GFP fluorescence with or without Arg-C treatment for 60 min. GFP signals were detected with a fluorescent microscope. The GFP signal was normalized with the chlorophyll autofluorescence signal. Error bars represent mean ± S.D. from independent experiments ($n = 3$). **$P < 0.01$, ***$P < 0.001$, two-sided student's $t$ test. Exact $P$ values are as follows, **$P = 0.00329$, ***$P = 0.00099$. Source data are provided as a Source Data file. **c** Western blotting analyses of AtVCCN1-mGFP and PsbO, with or without Arg-C treatment. AtVCCN1-mGFP and PsbO were immunoblotted by an anti-GFP antibody and an anti-PsbO antibody, respectively. The experiment was repeated three times independently, with similar results.

To further investigate the physiological properties of VCCN1, we evaluated its topology in the thylakoid membrane. The 'positive-inside rule' is also applicable to thylakoid membrane proteins: positively charged residues are more likely to be located within the stromal side of thylakoids (corresponding to the cytoplasmic side in cellular membranes)[22]. Based on this rule, the C-terminus of VCCN1 is predicted to be in the stroma. To experimentally prove this prediction, we conducted a protein digestion assay with C-terminally monomeric GFP-fused AtVCCN1 (AtVCCN1-mGFP), using chloroplasts isolated from *Arabidopsis thaliana*. The fluorescence of the AtVCCN1-mGFP was quenched upon the treatment with endoproteinase Arg-C, whereas such quenching was not observed without the proteinase treatment (Fig. 2a, b). Western blotting analyses confirmed that this quench is due to mGFP digestion, whereas a thylakoid protein, PsbO, one of the major components in the PSII core complex, was not digested by this treatment (Fig. 2c). These results demonstrated that the C-terminus of VCCN1 is exposed to the stroma, thus supporting our predicted topology in chloroplasts.

We then performed single-particle cryo-EM analyses of MdVCCN1 in detergent micelles (glyco-diosgenin; GDN), and in nanodiscs to investigate the structure in a native-like membrane environment (Supplementary Figs. 3–5). The structures were determined at 3.0 Å and 2.7 Å, respectively, with imposed C5 symmetry (Fig. 3a and Supplementary Fig. 4a). In both structures, all transmembrane (TM) and stromal helices showed clear densities (Supplementary Figs. 4f and 5f), which allowed the modeling of the MdVCCN1 structures, using the structure of the previously determined bacterial homolog from *Klebsiella pneumoniae* (KpBEST; PDB: 4WD8), with amino acids 84 to 419 in GDN and 78 to 418 in nanodiscs (Fig. 3b and Supplementary Fig. 6a). Both structures are basically similar, with root mean square deviation (RMSD) values below 0.52 Å for 329 C$_\alpha$ atoms between monomers (Supplementary Fig. 6b). Since the structure reconstituted in nanodiscs was determined at a higher resolution, we mainly describe this structure unless noted otherwise.

**Overall structure**. The folding and subunit assembly of MdVCCN1 are consistent with those of the two previously determined bestrophin channel structures from chicken Bestrophin 1 (GgBEST1; the "Gg" refers to *Gallus gallus*) and bovine Bestrophin 2 (BtBEST2; the "Bt" refers to *Bos taurus*), as well as

the bacterial homolog structure[18–20,23] (Fig. 3b–e). Since GgBEST1 is the best characterized channel, in terms of the structural and functional aspects among the previously studied species, we compared MdVCCN1 with GgBEST1. MdVCCN1 forms a symmetric homopentamer, with each subunit consisting of three domains: transmembrane domain (TMD), N-terminal domain (NTD), and stromal domain (SMD) (Fig. 3b). The TMD is composed of four transmembrane (TM) helices. TM1 and TM2 are located on the outer and inner sides of the transmembrane region, respectively, and TM3 and TM4 are sandwiched by these helices (Fig. 3b, e). TM2 constitutes the pentameric interface, and thus forms a central ion permeation pore. The NTD is composed of an N-terminal amphipathic helix (NAH), an N-terminal charged helix (NCH), and two short stromal helices (S1 and S2). The NAH and S2 helices are amphipathic and run parallel to the membrane, while the NCH and S1 helices are located on the stromal side (Fig. 3c, d). The NAH and NCH helices are only observed in VCCN1, whereas the S1 and S2 helices are also conserved in bestrophin channels. The SMD is composed of the C-terminal half of the long TM2 helix and six stromal helices (S3–S8). In the previously determined GgBEST1 structures, the long loop between the TM4 and S8 helices contains a cluster of acidic residues and constitutes the Ca$^{2+}$ clasp, which plays an essential role in the Ca$^{2+}$-dependent activation of bestrophin channels[24] (Supplementary Figs. 7 and 8). By contrast, the corresponding loop of MdVCCN1 lacks these acidic residues (Supplementary Fig. 7). The C-terminus of the bestrophin channel forms a long loop (> 200 amino acids) that plays an essential role in channel inactivation, by allosterically interacting with the cytosolic surface[25]. By contrast, the C-terminus of VCCN1 is composed of approximate 20 amino acids (Supplementary Fig. 7), and the densities of only 3 residues (Ala416-Gly418) were observed in the current cryo-EM maps, indicating that the C-terminus of MdVCCN1 is not involved in the channel regulation observed in bestrophin channels.

**Ion permeation pathway**. The ion permeation pathway of MdVCCN1 is formed by the TM2 and S7 helices from each subunit and located along the central axis, perpendicular to the membrane plane (Fig. 4a). The present structure revealed two constrictions along the pathway: one is located at the membrane region and the other is at the stromal end (Fig. 4a and Supplementary Fig. 6c). These constrictions at the membrane and the stromal end are conserved in other homolog structures and called

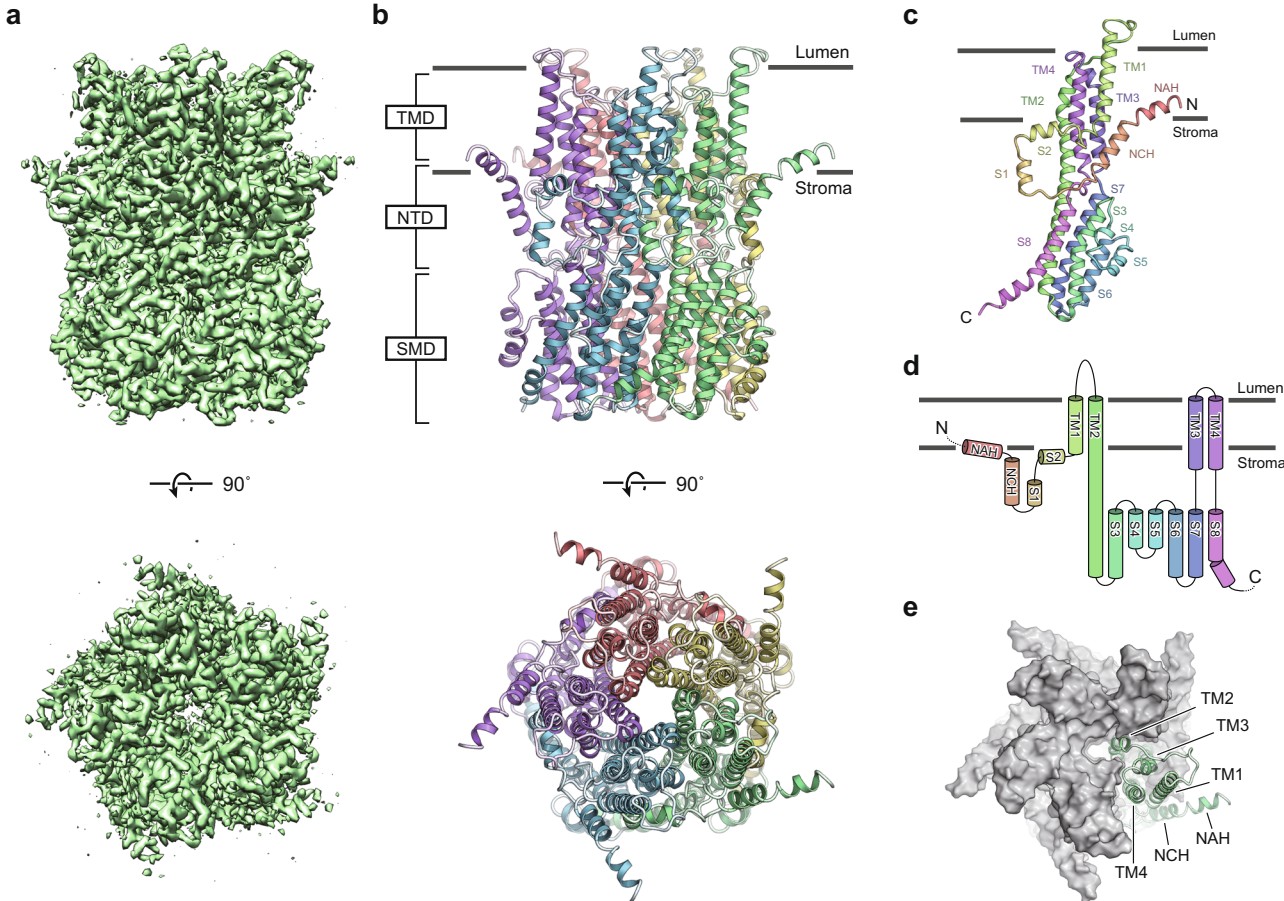

**Fig. 3 Overall structure of MdVCCN1. a** Cryo-EM density maps of MdVCCN1 reconstituted in nanodiscs, viewed from the side (upper panel) and from the top (lower panel). **b** Ribbon model of MdVCCN1 reconstituted in nanodiscs, viewed from the side (upper panel) and from the top (lower panel). The model is colored according to the chain. Gray bars represent the border between the thylakoid membrane and the solution. The predicted topology is indicated for VCCN1, based on the homologous bestrophin channels. **c** Structure of the MdVCCN1 protomer, colored by the helix. The color of each caption corresponds to each segment. **d** Schematic diagram of the MdVCCN1 topology, colored as in **c. e** Arrangement of TM helices of MdVCCN1, viewed from the top. Gray surface represents the molecular surface of four protomers.

the "neck" and the "aperture", respectively[18–20,23]. A comparison between the structures of the closed and $Ca^{2+}$-bound open GgBEST1 states revealed that the neck widens upon channel activation, while the aperture lacks significant conformational changes, demonstrating that the neck is mainly responsible for channel gating[20]. In the GgBEST1 structure, the neck is formed by a set of three hydrophobic residues (Ile76, Phe80, and Phe84) (Fig. 4b)[19,20]. Similarly, in the MdVCCN1 structure, the neck is composed of a set of three hydrophobic residues (Pro184, Leu188, and Phe192) (Fig. 4a, b and Supplementary Fig. 2) and constitutes the narrowest constriction along the pore, with a 1.40 Å radius, which is too narrow to allow the permeation of dehydrated $Cl^-$ (1.81 Å) (Fig. 4a). Therefore, the present MdVCCN1 structure represents the closed state.

The aperture of MdVCCN1 is formed by Ser301 from each subunit (Fig. 4c and Supplementary Fig. 2). In GgBEST1, Val205 constitutes the aperture and its side chain blocks hydrated $Cl^-$ permeation[24] (Fig. 4a, c). By contrast, MdVCCN1 has a wider aperture (3.30 Å), and the most constricted point is replaced by a smaller residue, Ser301. Thus, hydrated $Cl^-$ should permeate through the aperture of MdVCCN1. To test this hypothesis, we measured the selectivity of MdVCCN1 for several monovalent anions. While GgBEST1 has a narrower aperture and permeates monovalent anions depending on the lyotropic series[20], the permeabilities of $Cl^-$, $Br^-$, $I^-$, $NO_3^-$, and $SCN^-$ by MdVCCN1

were essentially the same or similar, except for $SCN^-$ (Fig. 4d–f). Overall, the aperture of MdVCCN1 does not seem to function as the selective filter, in contrast to GgBEST1. It should be noted that the GgBEST1 mutant with a wider aperture shows less selective permeability even between $SCN^-$ and other anions[20], suggesting the distinct mechanisms of ion selectivity between VCCN1 and BEST1.

**Subunit interface**. While the subunit interface is mainly formed by hydrophobic interactions through TM2, TM3, and S7, a pair of non-hydrophobic residues, Glu195 in TM2 and Tyr332 in TM3, forms a direct hydrogen bond in the transmembrane region between the adjacent subunits (Fig. 5a). This interaction is located at one helical turn below Phe192 of the neck region, as viewed from the luminal side, and apart from the central axis. These two residues are highly conserved among the VCCN1 proteins, suggesting the importance of this hydrogen bond (Supplementary Fig. 2). Accordingly, we hypothesized that this hydrogen bond contributes to maintaining the structural integrity of the neck region and thereby the proper channel gating.

To test our hypothesis, we created a series of MdVCCN1 Glu195 (E195A/D/L/Q/R) and Tyr332 (Y332A and Y332F) mutants, and performed whole-cell patch-clamp recordings. All of the tested Glu195 mutants exhibited whole-cell currents reduced by one-half

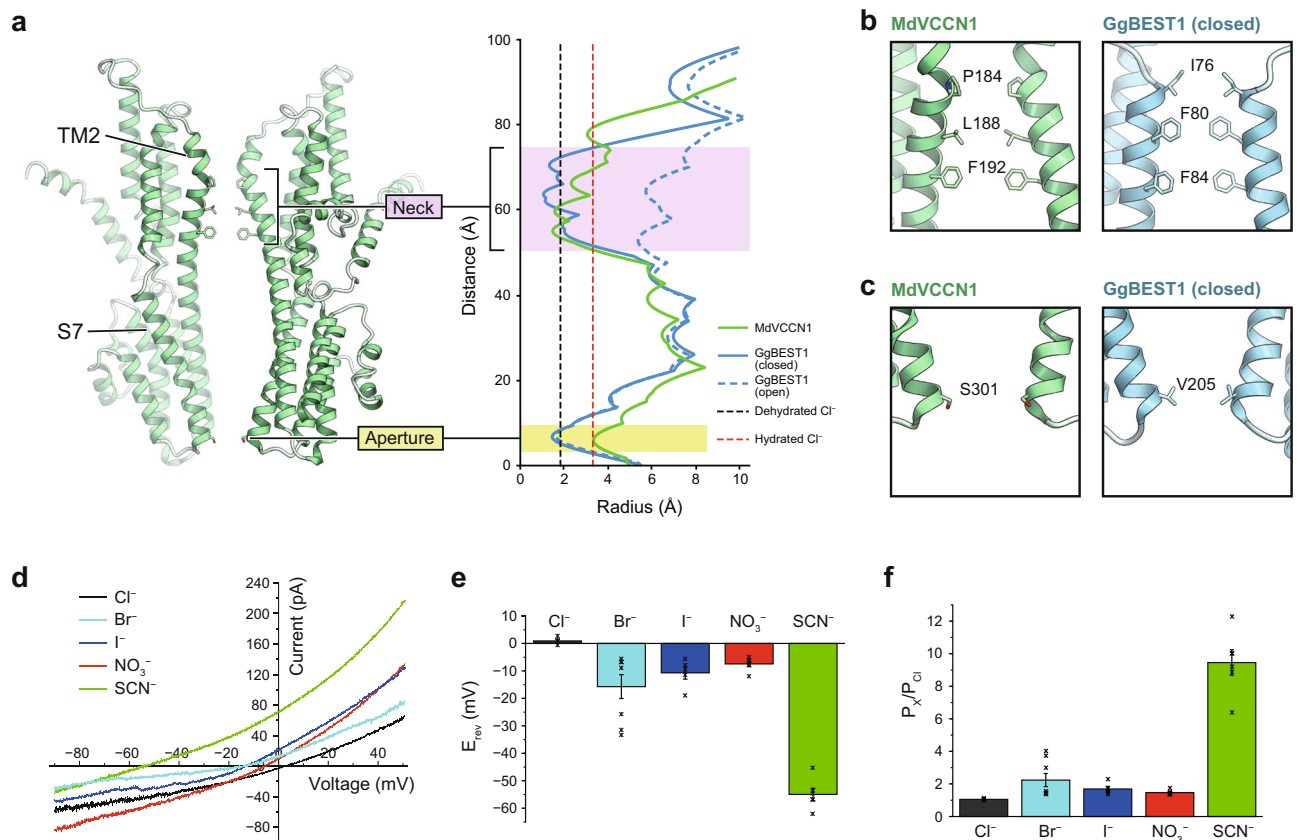

**Fig. 4 Ion permeation pathway and anion selectivity of MdVCCN1. a** Overview of the ion permeation pathway of MdVCCN1, and pore radii of MdVCCN1 (green) and GgBEST1 (blue: solid line represents closed state (PDB: 6N23) and dashed line represents open state (PDB: 6N28)) along the center axis of the channels. Black and red dashed lines represent the radii of dehydrated and hydrated Cl⁻, respectively. **b** Residues forming the neck of MdVCCN1 (left panel) and the closed GgBEST1 (right panel). **c** Residues forming the aperture of MdVCCN1 (left panel) and the closed GgBEST1 (right panel). **d, e** Representative traces of whole-cell current-voltage relationships (**d**) and reversal potentials (**e**) for the presence of NaCl ($n = 4$), NaBr ($n = 8$), NaI ($n = 5$), NaNO₃ ($n = 6$), or NaSCN ($n = 6$) in the bath solution. In **e** error bars represent s.e.m. Source data are provided as a Source Data file. **f** Comparisons of the permeabilities of Br⁻ ($n = 8$), I⁻ ($n = 5$), NO₃⁻ ($n = 6$) and SCN⁻ ($n = 6$) to Cl⁻ ($n = 4$). Error bars represent s.e.m. Source data are provided as a Source Data file.

or more (Fig. 5b and Supplementary Fig. 9a–e). However, the Y332F mutant exhibited greatly reduced whole-cell currents, whereas the Y332A mutant showed only slightly reduced currents (Fig. 5b and Supplementary Fig. 9f, g). We also performed an FSEC analysis for all mutants to check their pentameric assembly (Supplementary Fig. 10a, b). A cell-surface biotinylation assay of E195A and Y332F, representative mutants with greatly reduced currents, was also performed to the check plasma-membrane targeting of the mutants (Supplementary Fig. 10d), and suggested that the decrease in the whole-cell current is due to their reduction in each channel activity. Taken together, while all of these mutations designed to disrupt the hydrogen bond between Glu195 and Tyr332 diminished the channel activity, the Y332A mutation exceptionally retains the comparable whole-cell current. The bulky sidechain of Tyr332, located at the interface of the two TM2 helices, is reminiscent of Trp287 of GgBEST1, which is known to be critical for the channel gating in GgBEST1[20]. By the structural analogy, we assumed that the conformational change of the Tyr332 residue triggers MdVCCN1 gating via the hydrogen bond interaction, and that the Y332A mutation may mimic the state after the conformational change. Accordingly, to test our hypothesis, we determined the structure of the MdVCCN1 Y332A mutant reconstituted in nanodiscs by cryo-EM (Supplementary Figs. 11 and 12). The overall assembly and subunit folding of the Y332A mutant are quite similar to those of the WT structure in nanodiscs, with RMSD values below 0.36 Å for 335 Cα atoms

(Fig. 5c), demonstrating that the Y332A mutation does not affect the overall structural integrity. In the mutant structure, however, the Glu195 residue flips toward the central axis, and the Phe192 residue in the neck flips away from the central axis and resides in the space occupied by Tyr332 in the WT structure (Fig. 5d, e and Supplementary Fig. 13). Consequently, Glu195 is directly exposed to the ion permeation pathway, and thus may affect the anion permeability. Notably, this local conformational change of the Phe residue was similarly observed in the open structure of GgBEST1[20], while the other neck residues remained in the closed conformation (Fig. 5d). Consistently, the pore radius of the Y332A mutant was widened only at the position of Phe192 (Fig. 5e). Thus, the Y332A mutant seems to mimic the 'partially opened' state of VCCN1, whereas the whole neck opening requires larger movements, triggered by the conformational change of Tyr332. Taken together, these electrophysiological and structural analyses demonstrated that the inter-helical interactions near the neck region are important for channel activity in the VCCN1 family.

**N-terminal extension.** On the luminal side of the MdVCCN1 structure, the N-terminal 40 residues are folded into two short α-helices, an N-terminal amphipathic helix (NAH) and an N-terminal charged helix (NCH), which is a unique feature in VCCN1 and not conserved in the previously determined homolog structures (Supplementary Fig. 7)[18–20,23]. The NAH runs parallel to the membrane,

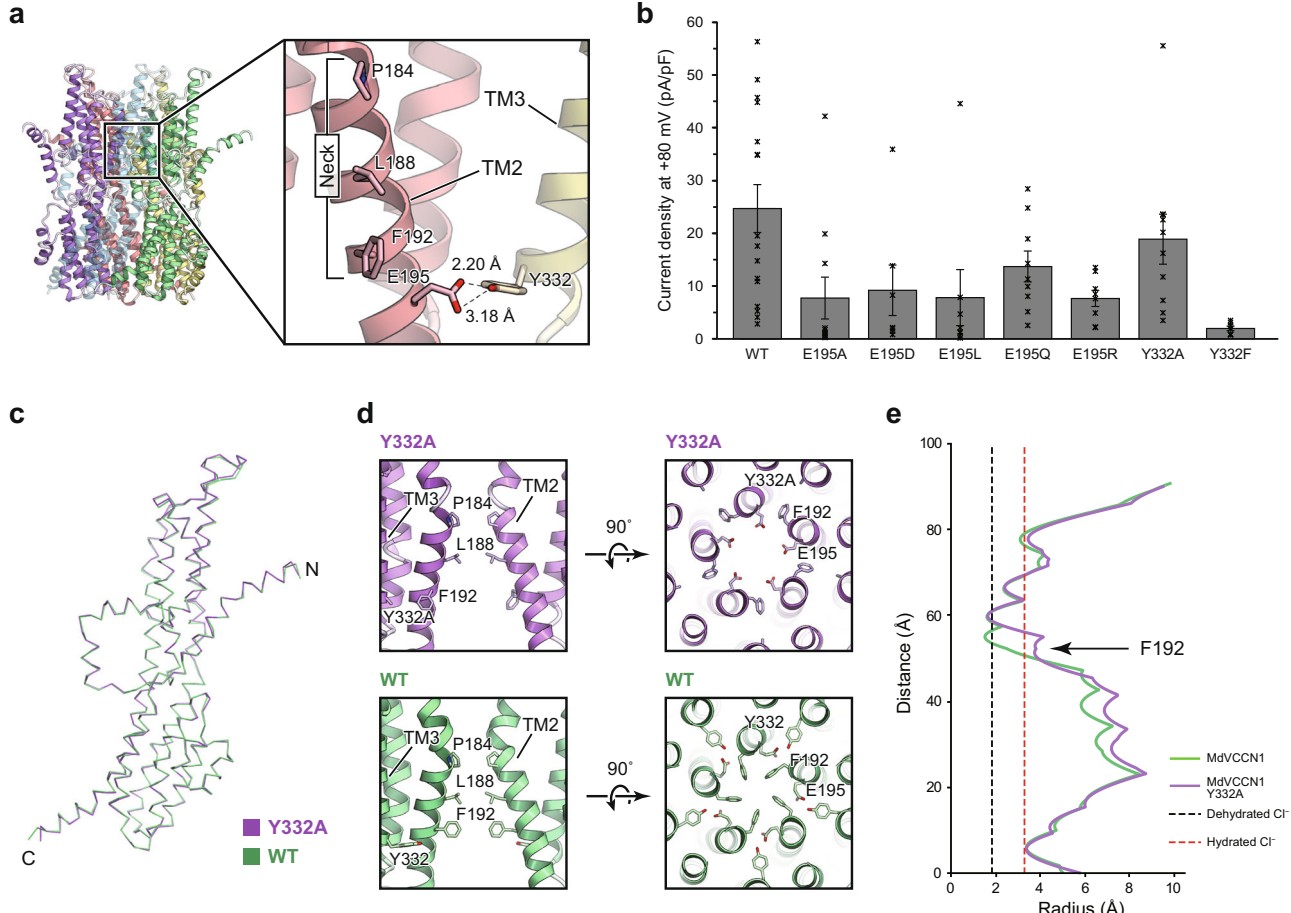

**Fig. 5 The hydrogen bond between Glu195 and Tyr332. a** Close-up view of the neck and the hydrogen bond between Glu195 and Tyr332. **b** Whole-cell current densities at +80 mV in HEK293 cells expressing MdVCCN1 wild-type, Glu195 mutant, or Tyr332 mutant (wild-type ($n = 16$), E195A ($n = 11$)], E195D ($n = 7$), E195L ($n = 8$), E195Q ($n = 9$), E195R ($n = 8$), Y332A ($n = 10$), Y332F ($n = 7$)). Error bars represent s.e.m. Source data are provided as a Source Data file. **c** Comparison of $C_\alpha$ traces of the MdVCCN1 Y332A mutant (violet) and wild-type (green). **d** Neck structures of the MdVCCN1 Y332A mutant (upper panels) and wild-type (lower panels), in which the left panels are views from the side and the right panels are from the top. **e** Pore radii of the MdVCCN1 Y332A mutant (violet) and wild-type (green) along the center axis of the channels. Black and red dashed lines represent the radii of dehydrated and hydrated Cl⁻, respectively.

with its hydrophobic side directly anchored to the membrane plane (Figs. 3c, d and 6a; Supplementary Fig. 14a). The NCH forms an intra-subunit interaction with the N-terminal tip of the TM1 helix and an inter-subunit interaction with the S1 helix of the adjacent subunit (Figs. 3c, d and 6a). Notably, the NAH is only visualized in the nanodisc structure and is disordered in the detergent micelles (Supplementary Fig. 14b), indicating that the lipid environment is essential to stabilize this helix.

To investigate the functional roles of NAH and NCH, we performed whole-cell patch-clamp recordings using various truncated or mutated constructs of the two helices (Supplementary Fig. 15). The N-terminal truncation mutant (ΔN; Pro69-Thr107 truncated), which lacks the NAH and NCH helices, showed significantly reduced current densities at +80 mV (Fig. 6b), while its expression in the plasma membrane was comparable to that of wild-type (Supplementary Fig. 10c, d), indicating the indispensable role of the NAH and NCH helices in the channel activity of MdVCCN1. To further investigate the functional role of the N-terminus, we substituted the acidic or basic residues on NCH with alanine or polar uncharged residues (D92A/D95A/E99A, D92N/D95N/E99Q, R88A/R96A/R100A/R103A, R88Q/R96Q/R100Q/R103Q) and measured their channel activities. All of the tested mutants reduced the current densities to levels similar to

or less than that of ΔN (Fig. 6b), indicating the essential role of the charged residues for the channel activity. Taken together, these results suggest that the electric charge distribution in NCH and its interaction with the lipid membrane through NAH contribute to the channel function of MdVCCN1.

## Discussion

In this study, we determined the cryo-EM structures of MdVCCN1, a plant homolog of bestrophin channels, in detergent micelles and in nanodiscs. As in other bestrophin family proteins, the overall structure exhibits a pentameric assembly with the ion permeation pathway along the central axis, which contains two constriction sites: neck and aperture. The neck of MdVCCN1 consists of three hydrophobic residues and is likely to restrict the ion permeation of MdVCCN1 in the closed state, similar to bestrophin channels. The other constriction site, the aperture, has a larger pore radius and thus does not function as an ion selective filter, in contrast to GgBEST1[24]. Previous studies on bestrophin channels revealed that mutations to enlarge the aperture resulted in reduced ion selectivity[20,24]. VCCN1 channels have distinct aperture residues (Supplementary Fig. 2), suggesting that the properties and functions of the apertures may vary among the species.

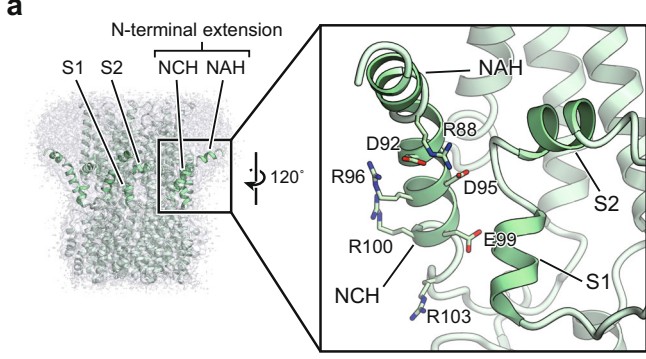

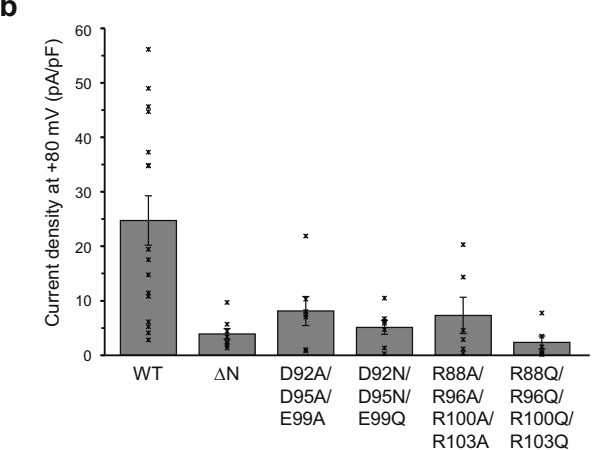

**Fig. 6 N-terminal extension of MdVCCN1. a** Overview and close-up view of the NTD of MdVCCN1. The gray density in the overview represents the cryo-EM map including the lipid nanodiscs. **b** Whole-cell current density at +80 mV in HEK293 cells expressing the MdVCCN1 wild-type, N-terminal truncated mutant and NCH mutant (wild-type ($n = 16$), ΔN ($n = 8$), D92A/ D95A/E99A ($n = 7$), D92N/D95N/E99Q ($n = 7$), R88A/R96A/R100A/ R103A ($n = 7$), R88Q/R96Q/R100Q/R103Q ($n = 6$)). Error bars represent s.e.m. Source data are provided as a Source Data file.

While the bestrophin channel is activated by $Ca^{2+}$, VCCN1 reportedly serves as a voltage-gated $Cl^-$ channel[14,15]. Consistently, our MdVCCN1 structure lacks the cluster of acidic amino acids that constitutes the $Ca^{2+}$-binding motif of bestrophin channels[24]. Instead, VCCN1 contains two N-terminal helices (NAH and NCH) that play crucial roles in the channel gating. Typical voltage-gated ion channels (VGIC) contain a cluster of positively charged residues on the transmembrane helix, known as the S4 segment, which serves as the voltage sensor[26,27]. The current structure revealed that VCCN1 has a similar cluster of charged residues on the NCH in the N-terminal extension, but it is located on the stromal side and not embedded in the membrane, unlike the S4 segment. This structural difference may explain the different voltage dependency profiles between VCCN1 and VGIC. VCCN1 was first identified as a voltage-gated anion channel in thylakoid membranes, based on the phenotypes of the mutants[14,15], but our electrophysiological experiments, as well as the previous research on AtVCCN1[15], revealed the much lower voltage-dependency of VCCN1, as compared to those of typical VGICs. Although this weak voltage-dependency might be partly due to our experimental environments, which may differ from thylakoid membranes in many aspects, it is conceivable that the voltage-gating profile of VCCN1 may be distinct from those of canonical VGICs. Furthermore, considering that the activation of VCCN1 is associated with the pH change in the chloroplast

lumen and that the NCH of VCCN1 is located in the stroma, VCCN1 might not be activated by the change of ΔΨ, but by other secondary signals on the stromal side derived from the light-induced PMF generation. While previous studies focused only on the physiological aspects of VCCN1, the current study for the first time has revealed the existence of the N-terminal extension, which is unique in VCCN1 and essential for channel function.

The structural comparison between MdVCCN1 and GgBEST1 revealed that the NCH and the following loop of MdVCCN1 are located at positions structurally similar to those of the $Ca^{2+}$ clasp and the C-terminal inactivation peptide in GgBEST1, on the stromal/cytosolic surface of the channels (Supplementary Figs. 8 and 16). These structural analogies suggest that the N-terminal extension of MdVCCN1 is involved in the channel regulation in VCCN1, which is consistent with our patch-clamp experiments on the mutants lacking these regulatory elements. As a result, the N-terminal extension enables VCCN1 to be activated by a distinct stimulation from that of bestrophin, but in a similar manner. In bestrophin channels, the structural changes in the $Ca^{2+}$ clasp are propagated to the neck region to open the channel pore[24]. Similarly, in VCCN1, the membrane voltage probably induces conformational changes in the NTD, which may be propagated to the neck region through inter-helical interactions, including the hydrogen bond between Glu on TM2 and Tyr on TM3. Although further functional and structural studies are required to clarify this gating mechanism, the present study provides the structural framework for understanding the molecular mechanisms of VCCN1, which plays a key role in photoreaction tuning. Recently, the detailed kinetics of VCCN1 in the photoreaction was investigated by using computer simulation[28]. This work suggested that further understanding of the NPQ regulation mechanisms and underlying kinetic features of ion channels and transporters in the thylakoid membrane can be applied to genetic engineering in plants, to optimize energy transduction in the photoreaction. Therefore, our findings help pave the way for such applications in bioenergetics.

## Methods

**Protein expression, purification, and sample preparation for cryo-EM.** The region encoding the N-terminally truncated MdVCCN1 protein (accession number XP_028961536.1), which lacks the N-terminal 67 amino acids (Leu2-Asp68) and contains one natural mutation (S207A), was amplified from a cDNA library prepared from *Malus domestica* leaves (ZYAGEN) and inserted into the pEG BacMam vector[29], with a C-terminal enhanced green fluorescent protein (eGFP)-His8 tag and a tobacco etch virus (TEV) protease cleavage site. Baculoviruses were generated in *Spodoptera frugiperda* Sf9 cells (American Type Culture Collection, catalog no. CRL-1711), using the Bac-to-Bac system (Invitrogen). HEK293S GnTI⁻ (N-acetylglucosaminyl-transferase I-negative) cells (American Type Culture Collection, catalog no. CRL-3022) were cultured in Freestyle 293 medium (Invitrogen) supplemented with 2% FBS (Sigma-Aldrich) in the presence of 8% $CO_2$, and infected by a P2 baculovirus at a density of approximately $3 \times 10^6$ cells ml⁻¹. After a 24 h incubation at 37 °C, the culture was supplemented with 5 mM sodium butyrate and incubated at 30 °C for 48 h. The cells were collected by centrifugation ($6,000 \times g$, 5 min, 4 °C) and disrupted by sonication in lysis buffer (50 mM Tris, pH 8.0, 150 mM NaCl, 10% glycerol). Cell debris was removed by centrifugation ($8,000 \times g$, 20 min, 4 °C). The membrane fraction was collected by ultracentrifugation ($186,000 \times g$, 1 h, 4 °C), and solubilized for 1 h at 4 °C in solubilization buffer (50 mM Tris, pH 8.0, 150 mM NaCl, 10% glycerol, 1% n-dodecyl-β-D-maltoside (DDM; Calbiochem)). Insoluble materials were removed by ultracentrifugation ($186,000 \times g$, 1 h, 4 °C). The detergent-soluble fraction was incubated with CNBr-Activated Sepharose 4 Fast Flow beads (GE Healthcare) coupled with an anti-GFP nanobody (GFP enhancer)[30] for 2.5 h at 4 °C. The beads were poured into an open column and washed with wash buffer (50 mM Tris, pH 8.0, 150 mM NaCl, 10% glycerol, 0.05% glyco-diosgenin (GDN; Anatrace)), and further incubated overnight with TEV protease to remove the eGFP-His8 tag. After the TEV protease digestion, the flow-through was collected, concentrated, and purified by size-exclusion chromatography (SEC) on a Superose 6 Increase 10/300 GL column (GE Healthcare), equilibrated with SEC buffer (50 mM Tris, pH 8.0, 150 mM NaCl, 0.05% GDN). The peak fractions were collected, concentrated to 4.5 mg ml⁻¹ with a centrifugal filter unit (Merck Millipore, 100 kDa molecular weight cutoff), and immediately used for cryo-EM grid preparation. The MdVCCN1 Y332A protein was purified by the same procedure.

**Nanodisc reconstitution and sample preparation for cryo-EM**. Membrane scaffold protein (MSP1E3D1) purification and protein reconstitution into nanodiscs were performed as previously described[31], with some modifications. L-α-phosphatidylcholine (Soy) (SoyPC) dissolved in chloroform (Avanti) was dried using a nitrogen stream and then further dried under vacuum desiccation overnight to remove the residual chloroform. Dried lipids were rehydrated in buffer (50 mM Tris, pH 8.0, 150 mM NaCl, 10% glycerol, 1% DDM). MdVCCN1, MSP1E3D1, and SoyPC were mixed at a molar ratio of 1:5:200 and incubated on ice for 1 h. Afterwards, 40 mg ml$^{-1}$ of Bio-Beads SM2 (Bio-Rad), prewashed with SEC2 buffer (50 mM Tris, pH 8.0, 150 mM NaCl), were added and the sample was incubated at 4 °C with gentle agitation to remove the detergents. The Bio-Beads were replaced with fresh ones after 2 h and the sample was incubated at 4 °C overnight, with gentle agitation. The Bio-Beads were removed by passage through a PolyPrep column (Bio-Rad), and the lysate was ultracentrifuged (106,400 × $g$, 20 min, 4 °C) before SEC. Ultracentrifuged samples were purified by size-exclusion chromatography on a Superdex 200 Increase 10/300 GL column (GE Healthcare), equilibrated with SEC2 buffer. The peak fractions were collected, concentrated to an absorbance ($A_{280}$) of 0.9–1.5 with a centrifugal filter unit (Merck Millipore, 100 kDa molecular weight cutoff), and immediately used for cryo-EM grid preparation. The MdVCCN1 Y332A protein was reconstituted into nanodiscs by the same procedure.

**EM image acquisition and data processing**. A 3 μl portion of each concentrated MdVCCN1 protein sample was applied to a glow-discharged Quantifoil R1.2/1.3 Cu/Rh 300 mesh or R1.2/1.3 Au 300 mesh grid (Quantifoil), blotted using a Vitrobot Mark IV (FEI) under 4 °C and 100% humidity conditions, and then frozen in liquid ethane. For MdVCCN1 in detergent, the grid images were obtained with a Titan Krios G3i microscope (Thermo Fisher Scientific) operated at 300 kV and recorded by a Falcon III direct electron detector (Thermo Fisher Scientific). A total of 1,161 movies were obtained in the electron counting mode, with a physical pixel size of 0.8346 Å pixel$^{-1}$. The data set was acquired with the EPU software, with a defocus range of −1.0 to −2.5 μm. Each image was dose-fractionated to 50 frames at a dose rate of 0.96 e$^-$ pixel$^{-1}$ per second, to accumulate a total dose of 50 e$^-$ Å$^{-2}$. The movie frames were aligned in 5 × 5 patches and dose weighted in MotionCor2[32], and defocus parameters were estimated using CTFFIND 4.1[33]. For MdVCCN1 WT and Y332A in nanodiscs, the grid images were obtained with a Titan Krios G3i microscope operated at 300 kV and recorded by a K3 direct electron detector (Gatan). A total of 2,271 movies for WT and 2,835 movies for Y332A were obtained in the electron counting mode, with a physical pixel size of 0.83 Å pixel$^{-1}$. The data set was acquired with the SerialEM software[34], with a defocus range of −0.7 to −1.9 μm for WT and −0.6 to −2.0 μm for Y332A. Each image was dose-fractionated to 48 frames at a dose rate of 15 e$^-$ pixel$^{-1}$ per second, to accumulate a total dose of 48 e$^-$ Å$^{-2}$. The movie frames were aligned in 4 × 4 patches and dose weighted using RELION 3.1[35], and defocus parameters were estimated using CTFFIND 4.1[33]. Further data processing was performed with RELION 3.0 or 3.1[35]. The figures depicting the density maps were prepared with UCSF Chimera (https://www.cgl.ucsf.edu/chimera/).

MdVCCN1 in detergent: First, Laplacian-of-Gaussian (LoG) based auto-picking was performed, and a total of 322,637 particles were extracted in 3.3384 Å pixel$^{-1}$. After two-dimensional classification, 176,803 good particles were three-dimensionally classified with C5 symmetry. Subsequently, 117,986 particles in the good class were re-extracted in the original pixel size of 1.30406 Å pixel$^{-1}$ and refined with C5 symmetry. The overall gold-standard resolution (FSC = 0.143) was 3.02 Å, with the local resolution in the core transmembrane region extending to 2.9 Å and that in the peripheral region extending to 3.9 Å (Supplementary Fig. 4).

MdVCCN1 in nanodiscs: Template-based auto-picking was performed with two-dimensional class averages of tens of thousands of particles picked by LoG-based auto-picking as templates. A total of 730,330 particles were extracted in 3.32 Å pixel$^{-1}$. After two-dimensional classification, 555,477 good particles were three-dimensionally classified with C5 symmetry, using the low-pass filtered map of MdVCCN1 in detergent as the initial model. Subsequently, 345,938 particles in the good class were re-extracted in the original pixel size of 1.328 Å pixel$^{-1}$ and refined with C5 symmetry. The overall gold-standard resolution (FSC = 0.143) was 2.69 Å, with the local resolution in the core transmembrane region extending to 2.7 Å and that in the peripheral region extending to 3.5 Å (Supplementary Fig. 5).

MdVCCN1 Y332A in nanodiscs: Template-based auto-picking was performed with two-dimensional class averages of tens of thousands of particles picked by LoG-based auto-picking as templates. A total of 1,494,710 particles were extracted in 3.32 Å pixel$^{-1}$. After three rounds of two-dimensional classification, 444,277 good particles were three-dimensionally classified with C5 symmetry, using the low-pass filtered map of MdVCCN1 reconstituted in nanodiscs as the initial model. Subsequently, 107,192 particles in the good class were re-extracted in the original pixel size of 1.328 Å pixel$^{-1}$ and refined with C5 symmetry. The overall gold-standard resolution (FSC = 0.143) was 2.69 Å, with the local resolution in the core transmembrane region extending to 2.7 Å and that in the peripheral region extending to 3.5 Å (Supplementary Fig. 12).

**Model building**. An initial model of MdVCCN1 was generated by creating a homology model based on the *Klebsiella pneumoniae* bestrophin homolog (PDB: 4WD8), using the Phyre2 server[36], and built into the density map in COOT[37]. The

initial model was refined using PHENIX with secondary structure restraints[38]. Finally, the model was refined by Refmac5[39] using Servalcat[40] under C5 symmetry constraints and secondary structure restraints prepared using ProSmart[41]. The van der Waals radii of the ion pathway were calculated using the HOLE program[42]. The figures depicting the molecular structures were prepared with CueMol (http://www.cuemol.org/).

**Patch-clamp**. For patch-clamp experiments, HEK293T cells (American Type Culture Collection, catalog no. CRL-11268) were cultured on poly-lysine treated coverglass dishes for 24 h, and then transfected with the receptor constructs, using Fugene 6 (Promega Co.). MdVCCN1 and the mutants were overexpressed by the pIRES2-AcGFP1 vector. Whole-cell patch-clamp recordings were then performed 36 h after transfection. The bath solution contained 140 mM NaCl, 5 mM KCl, 2 mM CaCl$_2$, 1 mM MgCl$_2$, 10 mM glucose and 10 mM HEPES, pH 7.4. Borosilicate pipettes (Harvard Apparatus), with a resistance of 5–8 MΩ, were filled with the pipette solution (148 mM CsCl, 0.5 mM CaCl$_2$, 2 mM MgCl$_2$, 2 mM EGTA, 10 mM NaCl and 10 mM HEPES, pH 7.2). For selectivity experiments, the NaCl was replaced by NaBr, NaI, NaNO$_3$, and NaSCN. The final concentration of DIDS was 0.5 mM in the bath solution. Currents were measured before and after the addition of DIDS. The voltage was clamped and currents were recorded using an Axopatch 200B amplifier (Axon CNS, Molecular Devices), coupled to an A/D converter (Axon CNS, Molecular Devices) and controlled by the pCLAMP10 software (Axon CNS, Molecular Devices). Currents were filtered at 1 kHz and sampled at 5 kHz.

**Transient expression of AtVCCN1-mGFP using the protoplast transfection assay**. Total RNA was extracted from *Arabidopsis thaliana* rosette leaves using Isogen II (Nippon Gene, Japan) and reverse transcribed to cDNA using a Prime-Script II 1st Strand cDNA Synthesis Kit (Takara Bio, Japan). The *AtVCCN1* gene was then amplified and fused with the *mGFP* gene. The *AtVCCN1-mGFP* genes were inserted between *Sal*I and *Sac*I sites of the pJD301 plasmid[43]. *Arabidopsis thaliana* wild type plants were grown in soil for 8 weeks in a growth chamber with an 8/16 h day and night cycle, at 23 °C and 60% relative humidity. Protoplasts derived from *Arabidopsis thaliana* mesophyll cells were prepared by using the Tape-*Arabidopsis* Sandwich method[44]. The protoplasts were then transfected by the TEAMP method[45] with slight modifications. Approximately 2 × 10$^5$ protoplasts in 0.5 ml MMg solution (4 mM MES-KOH, pH 5.7, 15 mM MgCl$_2$, 400 mM mannitol) were mixed with 100 μg of plasmid DNA at room temperature. An equal volume of PEG–calcium transfection solution (35% PEG4000, 100 mM CaCl$_2$, 200 mM mannitol) was added gently and incubated at room temperature for 15 min. Afterwards, 1 ml of W5 solution (2 mM MES-KOH, pH 5.7, 154 mM NaCl, 125 mM CaCl$_2$, 5 mM KCl) was added gently and mixed. Protoplasts were centrifuged at 200 × $g$ for 30 s and the supernatant was removed. Protoplasts were resuspended gently in WI solution (4 mM MES-KOH, pH 5.7, 20 mM KCl, 500 mM mannitol). A 500 μl aliquot of the protoplast suspension was dispensed into each well of a 12 well plate and incubated at 23 °C for 16 to 20 h.

**Protease protection assay**. Protoplasts were osmotically disrupted by suspension in low osmotic buffer (50 mM HEPES pH7.6, 7 mM MgCl$_2$, 1 mM MnCl$_2$, 2 mM EDTA, 30 mM KCl, 0.25 mM KH$_2$PO$_4$, 20 mM DTT, 0.2% BSA) to isolate chloroplasts. The protease protection assay was performed by treating the chloroplasts with arginylendopeptidase for 60 min. To terminate the reaction, 20 mM PMSF was added. The GFP signals of the treated chloroplasts were observed with a fluorescence microscope BZ-X810 (Keyence, Japan). Image analysis was performed using the ImageJ software (https://imagej.nih.gov/ij/). Samples were denatured by adding Laemmli's sample buffer and 2 μg of chloroplast protein was subjected to 15% SDS-PAGE followed by a western blot analysis. The anti-GFP antibody (MBL, Japan), anti-PsbO antibody (Agrisera, Sweden) and anti-Rabbit IgG antibody (MBL, Japan) were used at dilutions of 1/3,000, 1/3,000, and 1/5,000 respectively.

**Cell surface biotinylation assay**. The cell surface biotinylation assay was performed as previously described[46], with slight modifications as follows. For transient transfections, HEK293T cells were grown in poly-L-lysine-coated 6-well plates to ~50% confluence, and then transfected with 1 μg plasmid and 3 μl Mirus TransIT-LT1 in 100 μl Opti-MEM. Two days after transfection, the cells were washed twice with ice-cold PBS and incubated with 1 ml PBS supplemented with 5 mg/ml sulfo-NHS-LC-biotin (Thermo Fisher Scientific) for 30 min at 4 °C. The cells were washed twice with ice-cold PBS and quenched by 50 mM NH$_4$Cl in PBS for 10 min at 4 °C. After another wash, the cells were lysed by 1 ml ice-cold lysis buffer (0.5% Triton X-100 and 1 mM EDTA in PBS) containing cOmplete protease inhibitor cocktail (Roche) and incubated on ice for 15 min. After centrifugation at 20,000 g for 15 min at 4 °C, ten per cent of the cleared lysate was saved for immunoblotting, and the rest was incubated with 30 μl of Pierce™ High Capacity Streptavidin Agarose beads for 1 h at 4 °C. The beads were washed five times with lysis buffer, incubated in SDS-PAGE sample buffer containing 10% β-mercaptoethanol at 95 °C for 5 min to dissociate the biotinylated cell surface proteins from the beads, and analyzed by immunoblotting using an anti-GFP antibody (Roche) at dilution of 1/2,000.

**Reporting summary**. Further information on research design is available in the Nature Research Reporting Summary linked to this article.

## Data availability

The data that support this study are available from the corresponding author upon reasonable request. The atomic coordinates have been deposited in the Protein Data Bank (PDB) under the accession numbers 7EK1 (MdVCCN1 in GDN), 7EK2 (MdVCCN1 in nanodiscs), and 7EK3 (MdVCCN1 Y332A in nanodiscs). Cryo-EM density maps have been deposited in the Electron Microscopy Data Bank (EMDB) under the accession numbers EMD-31165 (MdVCCN1 in GDN), EMD-31166 (MdVCCN1 in nanodiscs), and EMD-31167 (MdVCCN1 Y332A in nanodiscs). Source Data are provided with this this paper.

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

## Acknowledgements

We thank R. Danev and M. Kikkawa for setting up the cryo-EM infrastructure, T. Nakane for computational support and assistance with the single-particle analysis and K. Ogomori for technical assistance. This work was supported by a MEXT Grant-in-Aid for Specially Promoted Research (grant no. 16H06294) and JST CREST program (grant no. 20344981) to O.N., JSPS KAKENHI (grant no. 20H03216) to T.N., a grant-in-aid from Institute for Fermentation Osaka (grant no. K-2018-009) to H.K., and the Platform Project for Supporting Drug Discovery and Life Science Research (Basis for Supporting Innovation Drug Discovery and Life Science Research (BINDS)) from the Japan Agency for Medical Research and Development (AMED) under grant no. JP19am01011115 (support number 1110).

## Author contributions

T.H. purified VCCN1 and performed the cryo-EM trial, determined the structure, and planned the mutational analyses, under the supervision of T. Kato, T.N., and O.N. K.K.,

T. Kusakizako, K.Y. and T.N. assisted with the EM image data collection and the data analyses. S.H. and H.K. performed the protease protection assays. T.S. performed the cell surface biotinylation assays. Y.F. assisted with the functional characterization. A.D.M. performed patch-clamp analyses. T.H., T. Kato, G.K., T.N., and O.N. wrote the manuscript, with feedback from all of the authors. T.N. and O.N. supervised the research.

## Competing interests

O.N. is a co-founder and scientific advisor for Curreio. The other authors declare no competing interests.
