## [Peer Review File · Nature Communications]

Cryo-EM structures of thylakoid-located voltage-dependent chloride channel VCCN1REVIEWER COMMENTS

Reviewer #1 (Remarks to the Author):

The authors describe the structure of VCCN1, a thylakoid-located chloride channel previously shown to be important for regulating the partitioning of pmf between the proton gradient and the membrane potential difference components across thylakoid membrane. The authors should better explain and underline the novelty and the importance of their work in the context of bestrophin-like channels and also in the context of photosynthesis.

While the structure resolution is well-performed and the authors highlight interesting structural aspects also by introducing structure-driven mutations and measuring electrophysiological activity, fundamental information is missing regarding the biochemical characterization of the preparation. Another issue is the lack of proper control experiments and the lack of the indication of the number of experiments (e.g. electrophysiology in Figures 3,4,5. How many times were these experiments performed? What is shown on the graphs? Mean values +/- SD or +/- SEM? In details, the following points have to be addressed:

The choice of determining the structure of VCCN1 from *Malus domestica* (rather than of the model plant *Arabidopsis*) should be briefly mentioned. If the authors tried to resolve the structure of this protein also from other sources but were not successful, this should be stated.

There is no biochemical characterization of this preparation, and no details are given about the rationale of using the protein expression method (baculovirus in HEK293 cells), about the expression level and how pure the preparation was. Determination and description of these basic characteristics (at least in the supplementary material) are fundamental for publishing. In any heterologous expression system, contamination by e.g. VDAC may occur. VDAC also forms DIDS-sensitive channels. The electrophysiological experiments shown here, where the authors express the protein in HEK293 to show that it works as a channel, do not prove that the protein purified from the HEK293 cells is indeed VCCN1. Biochemical data on purification must be included.

Figure 1. shows only the sensitivity of the channel to DIDS. This experiment alone does not prove that the activity measured is that of VCCN1. Subsequent structure-function relationship studies carried out by the authors indicate that indeed they work with VCCN1, but more characterization should be added to Figure 1.

Line 154: how do the authors know that a part of the channel is located towards the stromal end? If the channel topology has been determined in the thylakoid either here or in other works, it must be mentioned. Indeed, topology determination is of utmost importance when interpreting structural data. Figures 4 and 5. The authors here report that various mutations, introduced on the basis of the structural information obtained here, exert an effect on the current density. However, there is a serious issue here. In particular, the decreased/altered current density may be due either to a decreased single channel conductance, or to an incomplete/impaired targeting of the channel to the cell membrane. Thus, the authors must show either the effect of the mutations at single channel effect for each introduced mutation, or the efficiency of channel protein targeting in HEK293 cells (mammalian cells used for plant protein expression) for each mutant form. Without this control, no reliable conclusion can be drawn.

Reviewer #2 (Remarks to the Author):

This manuscript of Hagino et al is about the EM-structures of MdVCCN1, a voltage gated Cl⁻ channel localized in the thylakoid membrane. They determined the structures of wildtype MdVCCN1 in the presence of GDN and nanodiscs at 3.0 Å and 2.7Å, respectively and of the Y332A mutant at 2.7Å. By combining the structures and electrophysiological results, they have shown the N-terminal sequence, which does not exist in homologous bestrophin structures, is important for the channel activity of

MdVCCN1. They also showed that an inter-subunit salt-bridge between E195 and Y332 is critical for selectivity filter formation at the "neck" region.

Major point

I found that the structural biology and electrophysiology experiments described in this manuscript are sound. This paper, however, provides little information to understand physiological role of this protein in the thylakoid membrane. In the introduction, they have stressed that the voltage-gating of this protein, which does not exist in the homologous bestrophins, is the key to prevent ROS formation from excessive photosynthetic activities. They, however, provides little structural basis of the voltage-gating in the manuscript. They have established functional assay of the protein therefore they should try to find the region critical for the voltage-gating using the structural information. They might have already tried but simply failed to find any critical regions. But, at least, they should discuss the critical regions for the gating using both structural biological and electrophysiological data. They should also provide the literature information about the gating mechanism.

Minor points

1. It is impossible to pick up transmembrane helices in the lower panel of fig. 2b thus not possible to compare with fig. 2e. Maybe, it is better to show the real structure of the transmembrane helices with numbers in the fig. 2e.
2. sFig. 6 should be improved. Only one residue is highlighted in the figure but all the residues and regions discussed in the manuscript should be highlighted. In addition, it helps to add a figure to compare the 3D structures of MdVCCN1 and GgBEST1.
3. Very minor. Fig. 3a and b. They are currently two separate panels but, indeed, are interconnected and it is not natural to separate it into two panels.
4. Very minor. line 175. "essentially equivalent" -> essentially the same or similar.
5. Subunit interface section. I do not understand the logic behind the authors chose the mildest mutant Y332A to study its structure. Authors should elaborate why they chose this mutant over more drastic ones. Is there any change of ion selectivity of this mutant from the wild-type?
6. N-terminal extension section. Electrophysiology clearly shows this region is important for the channel activity but are there any known physiological function of this region? In general, this paper poorly connects the experimental results to the physiological functions of this protein.

Our point-by-point responses to the reviewers' comments are as follows.

Responses to Referees

Reviewer #1:

The authors describe the structure of VCCN1, a thylakoid-located chloride channel previously shown to be important for regulating the partitioning of pmf between the proton gradient and the membrane potential difference components across thylakoid membrane. The authors should better explain and underline the novelty and the importance of their work in the context of bestrophin-like channels and also in the context of photosynthesis.

Answer:

Thank you for the kind review and critical comments on our manuscript. We apologize for our inadequate explanations of the novelty and the importance of this study.

In the context of bestrophin-like channels, we newly discovered that the VCCN1 channel is regulated by a helix located on the stromal surface (termed "NCH" in this paper), which is different from Ca²⁺-regulation in bestrophin channels. Our VCCN1 structure

revealed that the location of the NCH is similar to that of the Ca^{2+} binding site (“ Ca^{2+} clasp”) of bestrophin, and accordingly this structural feature probably enables VCCN1 to be activated by distinct stimulation. In order to emphasize this point, we revised the discussion section, in lines 287-290 and 297-299.

In the context of photosynthesis, our results suggested that the voltage sensing by VCCN1 is distinct from that of typical voltage-gated ion channels. Although VCCN1 was identified as voltage-gated anion channel, based on the phenotypes of the mutants, its voltage-dependency is much lower in our electrophysiological experiment, as compared to typical voltage-gated ion channels. Consistently, while we identified the NCH as the essential element for the channel activity of VCCN1, it is located on the stromal side and not embedded in the lipid membrane, and thus is structurally distinct from canonical voltage-sensors in voltage-gated channels. These differences even suggest the possibility that VCCN1 is not directly activated by the change of $\Delta\Psi$, but by other secondary signals derived from the light-induced PMF generation. These explanations were added in the discussion section, lines 271-287. It also should be mentioned that further understanding of the activation mechanism of VCCN1 is featured in terms of the application to genetic engineering in plants to optimize energy transduction in the photoreaction (Li *et al.*, *Nat. Plants*, 2021). We emphasized this important point in the discussion section, lines 306-311.

While the structure resolution is well-performed and the authors highlight interesting structural aspects also by introducing structure-driven mutations and measuring electrophysiological activity, fundamental information is missing regarding the biochemical characterization of the preparation. Another issue is the lack of proper control experiments and the lack of the indication of the number of experiments (e.g. electrophysiology in Figures 3,4,5. How many times were these experiments performed? What is shown on the graphs? Mean values +/- SD or +/- SEM? In details, the following points have to be addressed:

Answer:

Thank you for this comment. We apologize for our insufficient explanation and lack of important information regarding the sample preparation and the electrophysiological experiment. According to the comment, we added panels showing the SEC chromatograms and the gel images of SDS-PAGE in Supplementary Figs. 3 and 10, which confirm that the VCCN1 protein was successfully purified and reconstituted in nanodiscs. We also recorded the currents from HEK293 cells transfected with empty vectors or without transfection as the negative controls. These results were added in Fig. 1, and confirm that the currents recorded

from HEK293 cells transfected with MdVCCN1 plasmids are indeed derived from VCCN1. We also added the number of experiments and the meanings of the error bars in the legends of Figs. 1, 3, 4, 5, s9, and s14.

The choice of determining the structure of VCCN1 from Malus domestica (rather than of the model plant Arabidopsis) should be briefly mentioned. If the authors tried to resolve the structure of this protein also from other sources but were not successful, this should be stated.

Answer:

Thank you for this comment. We mentioned that VCCN1 from *Malus domestica* is most suitable for cryo-EM due to its monodisperse FSEC profile and high expression among the species tested, in lines 99-103. The chromatograms shown below (Cover Letter Figure 1) are the expression profiles of VCCN1 homologues confirmed by FSEC. Although we also tried to clone the VCCN1 gene from an *Arabidopsis thaliana* cDNA library, we failed to obtain the gene. Accordingly, we only tested MdVCCN1 for the cryo-EM analysis.

Cover Letter Figure 1 | FSEC Chromatograms of VCCN1 homologues. FSEC chromatograms of VCCN1 from apple (*Malus domestica*), orange (*Citrus sinensis*), tomato (*Solanum lycopersicum*), and corn (*Zea mays*). FSEC experiments were conducted on a Superdex 200 Increase 10/300 column.

There is no biochemical characterization of this preparation, and no details are given about the rationale of using the protein expression method (baculovirus in HEK293 cells), about the expression level and how pure the preparation was. Determination and description of these basic characteristics (at least in the supplementary material) are fundamental for publishing. In any heterologous expression system, contamination by e.g. VDAC may occur. VDAC also forms DIDS-sensitive channels. The electrophysiological experiments shown here, where the authors express the protein in HEK293 to show that it works as a channel, do not prove that

the protein purified from the HEK293 cells is indeed VCCN1. Biochemical data on purification must be included.

Answer:

Thank you for this comment. As mentioned above, to show the purity of the protein sample, we added the SEC chromatograms and the SDS-PAGE gel images as Supplementary Figs. 3 and 10. The most prominent peak in the SEC analysis and the band around 37 kDa in the SDS-PAGE correspond to the MdVCCN1 protein (about 41 kDa), whereas the band around 25 kDa corresponds to MSP1E3D1 (about 32 kDa).

The baculovirus expression system in combination with a mammalian promoter is called the Bacmam system (Goehring *et al.*, *Nat. Protoc.*, 2014) and commonly used for the structural studies of higher eukaryotic membrane proteins. This strategy was also employed in structural studies of plant membrane proteins such as *Arabidopsis thaliana* TPC1 (Jiang *et al.*, *Nature*, 2016), so we considered that an explanation of the reason why we used this strategy is not necessary, since it is broadly accepted in the structural biology field.

Regarding the electrophysiological experiments, we expressed VCCN1 homologues in plasma membranes of HEK293 cells, by removing their signal peptides, and measured their currents by whole-cell patch-clamp recordings. VDAC showed very strong currents, but it is localized in mitochondrial outer membranes. Therefore, given the absence of significant currents for the negative controls (added in Fig. 1a as mentioned above), it is unlikely that the measured current is from VDAC. We should also mention that our work is not the first case of the use of mammalian cells for functional analyses of plant ion channels expressed in organelles (e.g. *Arabidopsis thaliana* TPC (Jiang *et al.*, *Nature*, 2016)).

Figure 1. shows only the sensitivity of the channel to DIDS. This experiment alone does not prove that the activity measured is that of VCCN1. Subsequent structure-function relationship studies carried out by the authors indicate that indeed they work with VCCN1, but more characterization should be added to Figure 1.

Answer:

Thank you for this comment. As mentioned above, we added the results of whole-cell patch-clamp experiments on HEK293 cells transfected with empty vectors or without transfection, which showed no significant currents, in Fig. 1a.

Line 154: how do the authors know that a part of the channel is located towards the stromal

end? If the channel topology has been determined in the thylakoid either here or in other works, it must be mentioned. Indeed, topology determination is of utmost importance when interpreting structural data.

Answer:

We apologize for our insufficient explanation. Although we did not experimentally confirm the VCCN1 topology, it is generally accepted that thylakoid membrane proteins have the same topology as their homologues in plasma membranes, with the stromal and luminal sides of thylakoids corresponding to the cytoplasmic and extracellular sides, respectively. Therefore, we assigned the VCCN1 topology based on that of the bestrophin channels. We explained this in the figure legends, and this “predicted topology” is indicated for the model.

Figures 4 and 5. The authors here report that various mutations, introduced on the basis of the structural information obtained here, exert an effect on the current density. However, there is a serious issue here. In particular, the decreased/altered current density may be due either to a decreased single channel conductance, or to an incomplete/impaired targeting of the channel to the cell membrane. Thus, the authors must show either the effect of the mutations at single channel effect for each introduced mutation, or the efficiency of channel protein targeting in HEK293 cells (mammalian cells used for plant protein expression) for each mutant form. Without this control, no reliable conclusion can be drawn.

Answer:

Thank you for this comment. The fluorescence images of the cells expressing wild-type and mutant MdVCCN1 (Cover Letter Figure 2) confirmed their expression in the plasma membrane. In addition, we confirmed the expression levels of the mutants by FSEC analyses, using the fluorescence of C-terminally-fused GFP (Cover Letter Figure 3). The FSEC chromatograms showed that each mutant has an expression level comparable to the wild-type and retains the pentameric assembly. These results confirm that the decreased conductance of each mutant is due to the decreased channel activity and not to other reasons, such as impaired targeting of the mutant channels to the membrane.

Cover Letter Figure 2 | Fluorescence images of the cells expressing MdVCCN1. Fluorescence images of the HEK293 cells expressing MdVCCN1 wild-type or mutants, which are fused with GFP at the C-terminus.

Cover Letter Figure 3 | FSEC Chromatograms of MdVCCN1 mutants. a-c, FSEC chromatograms of MdVCCN1 wild-type and Glu195 mutants (a), Tyr332 mutants (b), and N-terminal mutants (c). FSEC analyses were conducted on a Superose 6 Increase 10/300 column.

Reviewer #2

This manuscript of Hagino et al is about the EM-structures of MdVCCN1, a voltage gated Cl-channel localized in the thylakoid membrane. They determined the structures of wildtype MdVCCN1 in the presence of GDN and nanodiscs at 3.0 Å and 2.7Å, respectively and of the Y332A mutant at 2.7Å. By combining the structures and electrophysiological results, they have shown the N-terminal sequence, which does not exist in homologous bestrophin structures, is important for the channel activity of MdVCCN1. They also showed that an inter-subunit salt-bridge between E195 and Y332 is critical for selectivity filter formation at the “neck” region.

Answer:

Thank you for the kind review. Our point-by-point answers to the comments are described below.

Major point

I found that the structural biology and electrophysiology experiments described in this manuscript are sound. This paper, however, provides little information to understand physiological role of this protein in the thylakoid membrane. In the introduction, they have stressed that the voltage-gating of this protein, which does not exist in the homologous bestrophins, is the key to prevent ROS formation from excessive photosynthetic activities. They, however, provides little structural basis of the voltage-gating in the manuscript. They have established functional assay of the protein therefore they should try to find the region critical for the voltage-gating using the structural information. They might have already tried but simply failed to find any critical regions. But, at least, they should discuss the critical regions for the gating using both structural biological and electrophysiological data. They should also provide the literature information about the gating mechanism.

Answer:

Thank you for mentioning these important points. According to the comment, we modified the discussion section to include more detailed explanations in lines 271-284. The points are as follows.

Regarding the voltage-gating, unlike the S4 helix of typical voltage-gated ion channels, VCCN1 does not contain such a cluster of basic residues in transmembrane region. In good accordance, the voltage-dependency of VCCN1 in our electrophysiological experiments is much lower than those of typical voltage-gated ion channels. This also occurred in the previous electrophysiological study using *Arabidopsis thaliana* VCCN1 reconstituted in lipid bilayers (Herdean *et al.*, *Nat. Commun.*, 2016). Thus, the gating mechanism of VCCN1 is distinct from that of the canonical voltage-dependent channels, and we could not identify the critical regions for the gating, such as the S4 helix.

We also tried to define the gating mechanism by analogy with bestrophin. The key steps upon channel gating in bestrophin are “Ca²⁺ binding to the Ca²⁺ clasp on the intracellular surface of the channel” and “neck opening at the channel pore” (Miller *et al.*, *eLife*, 2019). Based on the structural analogy between VCCN1 and bestrophin, we assumed that the NCH may sense the voltage change in VCCN1, corresponding to the Ca²⁺ binding in bestrophin, and thus lead to the neck opening in a similar manner to bestrophin. However, even in bestrophin, the manner by which the Ca²⁺ binding is structurally connected to the channel opening in the TM region remains poorly understood. Thus, we could not find the critical regions for the voltage-gating by structural homology.

Furthermore, we tried to perform mutation analyses to confirm the role of the NTD in the signal sensing and propagation, but the mutations of Arg118, Ser119, and Tyr121,

located in the NTD and directly interacting with the acidic residues of the NCH, severely affected the pentamer formation of VCCN1, as shown in the FSEC chromatogram below (Cover Letter Figure 4). Thus, we could not clarify the overall contribution of the NTD.

Cover Letter Figure 4 | FSEC Chromatograms of MdVCCN1 wild-type and NTD mutant. FSEC chromatograms of MdVCCN1 wild-type and R118A/S119A/Y121F mutant. FSEC analyses were conducted on a Superose 6 Increase 10/300 column.

Minor points

1. It is impossible to pick up transmembrane helices in the lower panel of fig. 2b thus not possible to compare with fig. 2e. Maybe, it is better to show the real structure of the transmembrane helices with numbers in the fig. 2e.

Answer:

Thank you for this comment. We modified Fig. 2e with the protein model view.

2. sFig. 6 should be improved. Only one residue is highlighted in the figure but all the residues and regions discussed in the manuscript should be highlighted. In addition, it helps to add a figure to compare the 3D structures of MdVCCN1 and GgBEST1.

Answer:

Thank you for this comment. According to the reviewer's suggestion, we highlighted the N-terminal charged residues of MdVCCN1, the inter-subunit hydrogen-bonding residues of MdVCCN1, and the Ca²⁺ clasp residues of GgBEST1 in the revised Supplementary Fig. 7. We also added the comparison of the 3D structures between MdVCCN1 and GgBEST1 as

Supplementary Fig. 8.

3. Very minor. Fig. 3a and b. They are currently two separate panels but, indeed, are interconnected and it is not natural to separate it into two panels.

Answer:

We agree with the comment, and accordingly we modified the panel numbers of Fig. 3.

4. Very minor. line 175. “essentially equivalent” -> essentially the same or similar.

Answer:

Thank you for this suggestion. We corrected the sentence.

5. Subunit interface section. I do not understand the logic behind the authors chose the mildest mutant Y332A to study its structure. Authors should elaborate why they chose this mutant over more drastic ones. Is there any change of ion selectivity of this mutant from the wild-type?

Answer:

We apologize for our poor explanation, and added more information in the subunit interface section, lines 200-207. We expected the Y332A mutant to adopt an open conformation, because the similar mutant of bestrophin, W287F, located between the TM2 helices similarly to Tyr332 of VCCN1, adopts the open conformation (Miller *et al.*, *eLife*, 2019). In VCCN1, we noticed the hydrogen bonding interactions by the TM2 and TM3 residues, and mutant analyses revealed their importance in the channel function. Among the mutants, Y332A retained a channel current comparable to the wild-type, and we thus expected to obtain an open conformation with this mutant. Contrary to our expectations, the Y332A structure still represents the closed pore, with a slightly widened pore radius, by inducing one of the neck residues, Phe192. Since the observed conformational change is reminiscent of the open conformation of the corresponding mutant in bestrophin, we supposed that “it mimics the ‘partially opened’ state of VCCN1 (line 220)”. We clearly explained those contexts and slightly toned down the explanation in the revised manuscript.

We did not confirm any change in the ion selectivity of the mutant. In the original manuscript, we concluded that “inter-helical interactions near the neck region are important

for **anion permeability** in the VCCN1 family”, but this might be misleading. We revised it as “..... are important for **channel activity** in the VCCN1 family” in line 224, as it only functions in channel gating and is probably not involved in anion selectivity.

6. N-terminal extension section. Electrophysiology clearly shows this region is important for the channel activity but are there any known physiological function of this region? In general, this paper poorly connects the experimental results to the physiological functions of this protein.

Answer:

Thank you for the comment. We enhanced our explanation in the discussion section, lines 287-290. While previous studies highlighted only the physiological aspects of VCCN1, the current study for the first time revealed the unique structural features of the N-terminal extension and its implications in the channel function. Such natural variants have not been reported yet. Therefore, how such mutations affect the phenotype will be a subject of future research.

Please address correspondence concerning this manuscript to Prof. Osamu Nureki, Department of Biological Sciences, Graduate School of Science, The University of Tokyo, 7-3-1 Hongo, Bunkyo-ku, Tokyo 113-0033, Japan. Phone: +81-3-5841-4392; Fax: +81-3-5841-8057. E-mail: nureki@bs.s.u-tokyo.ac.jp.

Sincerely yours,

Osamu Nureki
Professor

REVIEWER COMMENTS

Reviewer #1 (Remarks to the Author):

The authors addressed some of my comments in the revised version. However, there are a few points that require further consideration, as the authors missed my points in some cases. It is important to resolve the issues listed below in order to render this important work solid from all points of view.

1) On my opinion the experimental proof of the topology of VCCN1 in the thylakoid membrane must be proven biochemically, since it is very important for assigning the correct location of the different domains in the channel and thus to understand its function in the context of photosynthesis regulation. Otherwise the discussion introduced in line 286 is an unjustified statement. Antibodies are available against VCCN1 or against a tag that can be fused to the channel in order to do this. The argumentation of the authors based on the topology of bestrophin channel in the plasma membrane of mammalian cells is very weak, since such extrapolation might be erroneous. There is no guarantee that the topology of a channel in the plasmamembrane of a mammalian cell is the same as the one of the same channel in plants, in the thylakoid membrane. Also because the protein targeting/import routes are completely different in the two systems.

2) The experiments shown only in the cover letter in response to my observations on Figures 4 and 5 are not convincing and acceptable, since the resolution and magnification of the images is not sufficient to prove that the WT and mutant channel proteins equally efficiently reached the plasma membrane. Actually it seems that in many cells there is intracellular retention of the proteins. The authors should have at least presented a co-localization study at higher magnification using a PM-specific dye. The FSEC Chromatograms of MdVCCN1 mutants, as presented, do not prove the PM localization of the channel, but give an information about the assembly, which may occur also in the ER: posttranslational folding, processing, and ultimately oligomerization of virtually all secretory pathway proteins occur in the ER, which provides "quality control" by identifying and degrading any misassembled proteins. For ion channels addition of auxiliary proteins may happen close to the PM or in the PM. In summary, FSEC Chromatograms should be performed in pure PM preparations, otherwise they do not prove correct PM targeting of VCCN1. Other possibilities to prove correct targeting to the PM are also possible, for example biotinylation studies, single channel patch clamp of the PM etc., at least for Glu195Ala and Tyr332Phe. This result, once obtained, must also be shown for the readers. It is important to give the correct message to the readers about the importance of the introduced mutations for VCCN1 function, namely it is mandatory to identify if these mutations indeed affect channel conductance or decrease the whole-cell current due to mistargeting of the proteins (as it seems from Fig 2 of the cover letter).

Reviewer #2 (Remarks to the Author):

In the revised manuscript, authors have fully addressed all my comments properly, thus this reviewer thinks that the manuscript is now ready for publication.

Our point-by-point responses to the reviewers' comments are as follows.

Responses to Referees

Reviewer #1:

The authors addressed some of my comments in the revised version. However, there are a few points that require further consideration, as the authors missed my points in some cases. It is important to resolve the issues listed below in order to render this important work solid from all points of view.

Answer:

Thank you for your kind review and critical comments on our revised manuscript. Our point-by-point answers to your comments are described below.

1) On my opinion the experimental proof of the topology of VCCN1 in the thylakoid membrane must be proven biochemically, since it is very important for assigning the correct location of the different domains in the channel and thus to understand its function in the context of photosynthesis regulation. Otherwise the discussion introduced in line 286 is an unjustified statement. Antibodies are available against VCCN1 or against a tag that can be fused to the channel in order to do this. The argumentation of the authors based on the topology of bestrophin channel in the plasma membrane of mammalian cells is very weak, since such extrapolation might be erroneous. There is no guarantee that the topology of a channel in the plasmamembrane of a mammalian cell is the same as the one of the same channel in plants, in the thylakoid membrane. Also because the protein targeting/import routes are completely different in the two systems.

Answer:

Thank you for this comment. Firstly, there is a consensus in the topology of membrane proteins, known as the "positive-inside rule". Based on this rule, positively charged residues are more likely to be located in the cytoplasmic region (von Heijne, 2006). This concept is also applied to thylakoid membrane proteins with the stromal and luminal sides of thylakoids corresponding to the cytoplasmic and extracellular sides, respectively (Gavel et al., 1991). Our prediction based on the structural homology with bestrophin is supported by this concept. Consistently, our MdVCCN1 structure shows that the molecular surface is positively charged on the predicted stromal side (Cover Letter Figure 1). We added the description of this rule to the "Functional characterization

and structure determination of MdVCCN1” section, lines 115-120.

Nonetheless, we agreed with the Reviewer’s comment that the experimental determination of the topology in native chloroplasts is important, and thus conducted a protein digestion assay using plant chloroplasts. We transiently expressed C-terminally GFP-fused AtVCCN1 in protoplasts derived from *Arabidopsis thaliana*. We then isolated the chloroplasts in a hypo-osmotic solution and treated them with endoproteinase Arg-C to check its sensitivity. Since the hypo-osmotically isolated chloroplast envelope exhibits very high permeability (Endo et al., 1998), Arg-C can access the thylakoid membrane without any treatment, such as the usage of detergent. In this assay, GFP fluorescence would be quenched by Arg-C digestion, if the C-terminal GFP is exposed outside the thylakoid membrane. We successfully observed Arg-C-dependent GFP quenching, and GFP digestion was further confirmed by a western blotting analysis (Cover Letter Figure 2). By contrast, Arg-C-dependent digestion was not observed in PsbO, one of the major components in the Photosystem II core complex located in the thylakoid lumen (Cover Letter Figure 2c). Considering that PsbO contains more Arg-C digestion sites than GFP (9 in PsbO and 6 in GFP), the different susceptibilities to Arg-C probably reflect the different topologies of the two proteins in thylakoids, thus demonstrating that the C-terminus of VCCN1 is exposed to the stroma, as in our model. We added these descriptions and results to the “Functional characterization and structure determination of MdVCCN1” section, lines 120-128. We also added the related figures to Fig. 2.

Cover Letter Figure 1 | Electrostatic potential of MdVCCN1. The molecular surface of MdVCCN1 colored according to the electrostatic potential, from negative in red to positive in blue.

Cover Letter Figure 2 | Topology evaluation of VCCN1 in chloroplasts. a, Representative images of GFP digestion by Arg-C on isolated chloroplasts with transient expression of AtVCCN1-mGFP. The scale bar represents 50 μm . **b,** Quenching level of GFP fluorescence with or without Arg-C treatment for 60 min. GFP signals were detected with a fluorescent microscope. The GFP signal was normalized with the chlorophyll autofluorescence signal. Error bars represent S.D. ($n = 3$). **, $p < 0.01$, ***, $p < 0.001$, student's t test. **c,** Western blotting analysis of AtVCCN1-mGFP and PsbO, with or without Arg-C treatment. AtVCCN1-mGFP and PsbO were immunoblotted with an anti-GFP antibody and an anti-PsbO antibody, respectively.

2) The experiments shown only in the cover letter in response to my observations on Figures 4 and 5 are not convincing and acceptable, since the resolution and magnification of the images is not sufficient to prove that the WT and mutant channel proteins equally efficiently reached the plasma membrane. Actually it seems that in many cells there is intracellular retention of the proteins. The authors should have at least presented a co-localization study at higher magnification using a PM-specific dye. The FSEC Chromatograms of MdVCCN1 mutants, as presented, do not prove the PM localization of the channel, but give an information about the assembly, which may occur also in the ER: posttranslational folding, processing, and ultimately oligomerization of virtually all secretory pathway proteins occur in the ER, which provides "quality control" by identifying and degrading any misassembled proteins. For ion channels addition of auxiliary proteins may happen close to the PM or in the PM. In summary, FSEC Chromatograms should be performed in pure PM preparations, otherwise they do not prove correct PM targeting of VCCN1. Other possibilities to prove correct targeting to the PM are also possible, for example biotinylation studies, single channel patch clamp of the PM etc., at least for Glu195Ala and Tyr332Phe. This result, once obtained, must also been shown for the readers. It is important to give the correct message to the readers about the importance of the

introduced mutations for VCCN1 function, namely it is mandatory to identify if these mutations indeed affect channel conductance or decrease the whole-cell current due to mistargeting of the proteins (as it seems from Fig 2 of the cover letter).

Answer:

Thank you for this comment. We used the cell-surface biotinylation assay in HEK293 cells to assess the wild-type and E195A, Y332F or Δ N mutants, which are the representative mutants with decreased whole-cell currents. All three mutants showed almost comparable levels of surface biotinylated proteins (Cover Letter Fig. 3). Therefore, the decreased whole-cell currents of these MdVCCN1 mutants are not due to their incorrect membrane targeting. We added these results in the “Subunit interface” section and modified the “N-terminal extension” section, lines 220-225 and 269. We also added the corresponding figure as Supplementary Fig. 10d.

Cover Letter Figure 3 | Cell surface biotinylation assay for MdVCCN1 mutants. Cell surface biotinylation assay for HEK293 cells expressing GFP-fused MdVCCN1 wild-type, E195A, Y332F, and Δ N mutants with controls (GFP only or empty vector transfected HEK cells). The samples were immunoblotted with an anti-GFP antibody. The band intensities of the pulled-down GFP-tagged proteins were quantified and normalized to the input lysates (PD/Input).

Reviewer #2:

In the revised manuscript, authors have fully addressed all my comments properly, thus this reviewer thinks that the manuscript is now ready for publication.

Answer:

Thank you for the kind review on our revised manuscript.

Thank you again for your thoughtful consideration, and we hope that the revised manuscript is now acceptable for publication in *Nature Communications*.

Sincerely,

Osamu

REVIEWERS' COMMENTS

Reviewer #1 (Remarks to the Author):

The authors did an excellent job and addressed all my concerns by adding new, convincing experimental results.